# A neuronal basis for fear discrimination in the lateral amygdala

Anna Grosso[1], Giulia Santoni[1], Eugenio Manassero[1], Annamaria Renna[1] & Benedetto Sacchetti[1,2]

In the presence of new stimuli, it is crucial for survival to react with defensive responses in the presence of stimuli that resemble threats but also to not react with defensive behavior in response to new harmless stimuli. Here, we show that in the presence of new uncertain stimuli with sensory features that produce an ambiguous interpretation, discriminative processes engage a subset of excitatory and inhibitory neurons within the lateral amygdala (LA) that are partially different from those engaged by fear processes. Inducing the pharmacogenetic deletion of this neuronal ensemble caused fear generalization but left anxiety-like response, fear memory and extinction processes intact. These data reveal that two opposite neuronal processes account for fear discrimination and generalization within the LA and suggest a potential pathophysiological mechanism for the impaired discrimination that characterizes fear-related disorders.

---

[1] Rita Levi-Montalcini Department of Neuroscience, University of Turin, I-10125 Turin, Italy. [2] National Institute of Neuroscience - Turin, I-10125 Turin, Italy. Correspondence and requests for materials should be addressed to B.S. (email: benedetto.sacchetti@unito.it)

Encountering a new stimulus or situation represents one of the major challenges for organisms; i.e., which sensory stimuli should be approached and which should be avoided? For survival, animals, including humans, cannot spend time evaluating all possible consequences of a new situation. They must be able to predict them while the situation is on-going. To do this, organisms compare information about the context in which the new experience is occurring with stored knowledge about past emotional experiences. When new stimuli are perceptually similar to those associated with danger in the past, organisms respond with defensive responses, such as avoidance or immobility. Conversely, greater dissimilarity yields different behaviors (e.g., curiosity and exploration)[1–3].

As a site essential for detecting threats, the amygdala is also important for the generalization of fear responses to new stimuli that resemble threats[4–9], and the inappropriate activation of amygdala neurons may cause inappropriate fear reactions to harmless stimuli[4,10–14]. In this framework, little is known regarding whether and how the amygdala participates in the evaluation of new stimuli that are different from threats. It has been proposed that increasing dissimilarity between new and threat stimuli decreases overall activity in the amygdala, thereby preventing inappropriate fear reactions[6,7,9]. However, recent studies have shown that the amygdala may also take part in learning safety[15–17] and extinguishing previously learned threat stimuli[4,18,19]. Therefore, the present study is aimed at investigating the neuronal mechanisms that are activated within the amygdala during the evaluation of new stimuli that may be perceptually similar to or different from previously learned threats. Here we show that a specific subset of neurons within lateral amygdala (LA) are activated when a new stimulus is evaluated as "harmless". These cells differ in part from those engaged by fearful stimuli, and the pharmacogenentic blockade of these neurons caused fear generalization.

## Results

**Different LA activation by uncertain stimuli presentation.** Rats were trained to associate a pure tone of a specific frequency (conditioned stimulus, CS, 1 kHz) with a painful unconditioned stimulus (a mild electric foot shock, US). We chose this conditioning procedure to a single type of auditory stimulus because it mimics real-life threatening experiences that occur without fine and prolonged discrimination[8,20]. This allows the investigation of the neural processes that occur when subjects are facing totally new stimuli that may or may not resemble the threatening event. One week after training, we monitored the rats' behavior when presented with the CS (Group 1) or with new tones of increasing different frequencies (3 kHz, Group 2; 7 kHz, Group 3; or 15 kHz, Group 4). Twenty min later, all groups were presented with the CS to test their fear memory retention (Fig. 1a). As control group we used naive rats. Half of naive animals were exposed to the 1 and 7 kHz tones only during the test trial, while the others were exposed to 1 kHz tone unaccompanied by any US during training and, 1 week later, to either the 1 and the 7 kHz tones during test. Since we did not detect any differences between the two groups (Supplementary Fig. 1), they were collected altogether ("naive").

Rats exhibited marked defensive behavior (i.e., a freezing response) to the CS (1 kHz, Group 1) and to a new tone with a closer frequency (3 kHz, Group 2). Conversely, Group 3 displayed an intermediate level of freezing to the 7 kHz tone, whereas Group 4 showed less freezing to the 15 kHz tone (Fig. 1b). The behavior of the rats within each group was similar except for Group 3, in which the perceptual features of the tone were interpreted in two opposite directions: half of the rats (5/9) showed a low level of freezing ("discriminators", D), while the

others (4/9) exhibited higher freezing response ("generalizers", G) (Fig. 1c). The threshold (~43%) for assigning each animal to the "discriminator" or "generalizer" group was estimated through an expectation–maximization (EM) algorithm which yielded the maximum likelihood estimates for fitting a Gaussian mixture model (GMM) (see Methods and Supplementary Fig. 2). All conditioned groups displayed a higher and comparable level of freezing response to the CS with respect to naive animals (Supplementary Fig. 3). In all animals, to analyze neuronal activation by new tones and the CS in the LA, we performed a cellular compartment analysis of temporal activity using fluorescent in situ hybridization (catFISH), a technique that allows the detection of neurons activated by two different events[21–23]. We analyzed the RNA expression of two activity-dependent genes, Arc/Arg 3.1 (Arc) and Homer 1a (H1a) in the same animal (Fig. 1d, e). Arc mRNA is detected in the nucleus 5–8 min after a salient event, while H1a mRNA is visualized in the nucleus 25–30 min afterwards[22,23] (Supplementary Fig. 4). In animals exposed to two behavioral epochs separated by 20 min, epoch 1 drives nuclear H1a, while epoch 2 induces nuclear Arc expression (Fig. 1d). Single Arc or H1a expression therefore reflects selectivity for one of the two events, while double-labeling demonstrates that the same neuron is engaged by both epochs[21–23].

In Group 1, during the presentation of the two stimuli (CS–CS) there were many neurons activated during both events (Fig. 1e, f). Similar results were observed in Group 2 (3 kHz-CS) and in generalizer animals in Group 3 (7 kHz-CS) (Fig. 1e, f). Hence, fear generalization to a new tone recruited the same neurons that were activated by the CS. Conversely, in the discriminator animals in Group 3 (7 kHz-CS), the presentation of the new tone and the CS induced a significant reduction in the percentage of neurons expressing both Arc and H1a nuclear mRNA (i.e., neurons that responded to both the CS and the new tone) and increased the percentage of cells that responded separately to the CS and, strikingly, to the new tone (Fig. 1e, f). These data reveal that the LA is activated during the presentation of both harmful or new stimuli evaluated as not dangerous. In particular, in the presence of new stimuli evaluated as "harmless", LA activity relies on neurons that are partially different from those activated by the CS and that are silent during threatening experiences. The two different subpopulations of neurons that were activated by threatening vs harmless stimuli were intermingled within LA (Fig. 1e).

We then examined Group 4 (15 kHz-CS) to determine whether a similar phenomenon also occurs in the case of tones markedly different from the CS. However, in this group, the presentation of the new tone recruited a very low percentage of LA neurons (Fig. 1e, f). Notably, the discriminator animals in Groups 3 and 4 displayed similar behaviors during either CS or harmless stimuli, but only discriminator animals in Group 3 showed enhanced neuronal activity to either CS or harmless stimuli. These data allowed us to exclude that the neuronal activity detected only in Group 3 was the mere consequence of the motor or emotional behaviors. Altogether, these data demonstrate that, when stimulus features are markedly different from the CS, LA activity is lower than in the other conditioned groups, as previously proposed[6,7]. Conversely, when stimulus features may lead to opposite interpretations (i.e., safe or dangerous), in the LA there are neurons that are activated if the stimulus is evaluated as harmless. In the latter case, our data also suggest that the global activity of the LA is high and comparable to that displayed during threat events (Fig. 1f). This hypothesis was also corroborated by the fact that Discriminator animals in the Group 3 had higher number of activated neurons during new tone presentations than naive animals (Fig. 1f).

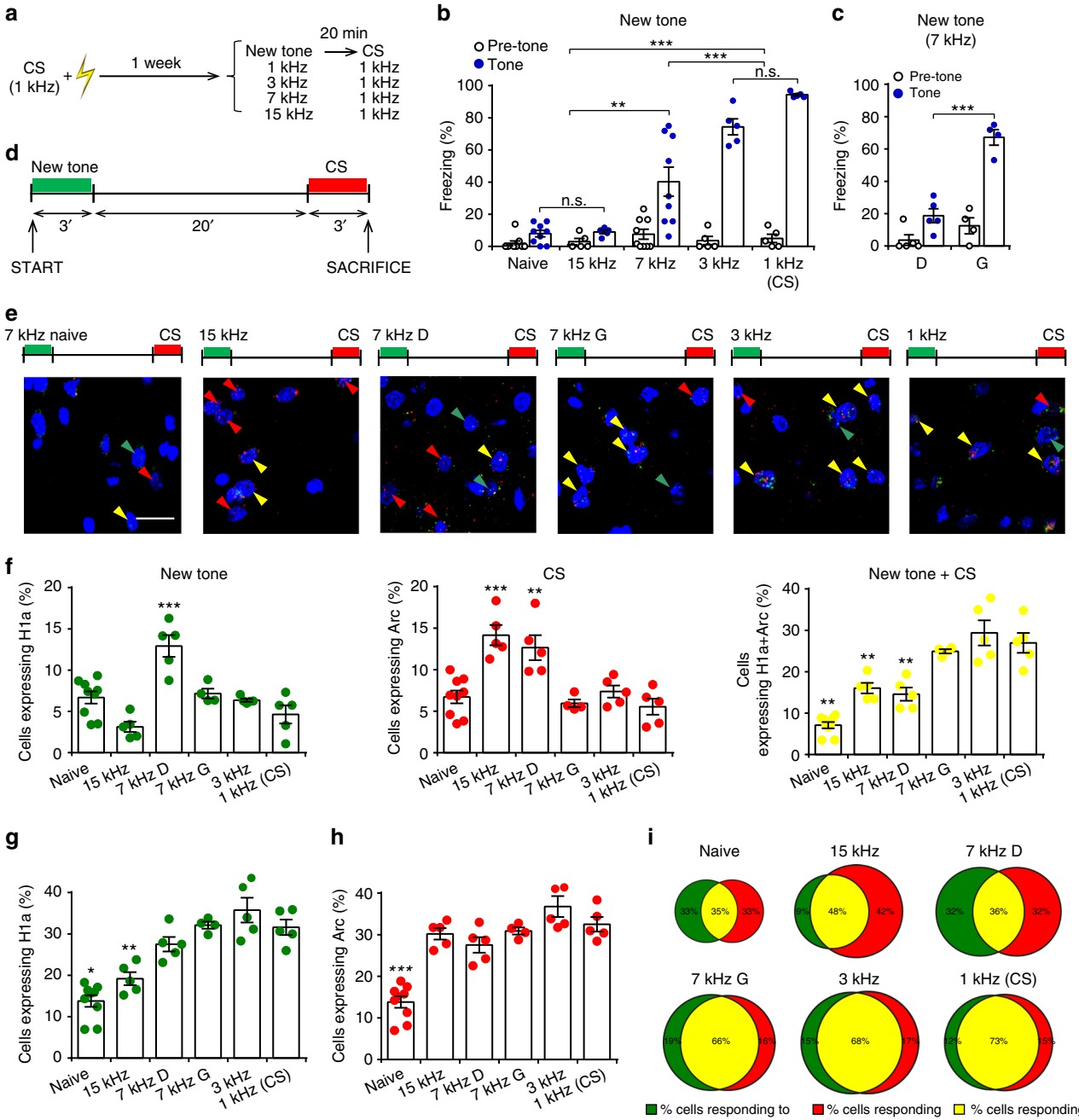

**Fig. 1** LA activity following the presentation of novel and threatening cues. **a** Experimental design of behavioral and catFISH experiments, see Methods. Naive $n = 9$, 15 kHz, $n = 5$; 7 kHz, $n = 9$; 3 kHz, $n = 5$; and 1 kHz, $n = 5$. CS conditioned stimulus. **b** The percentage of freezing following the new tone presentation progressively increased in similarity to the CS ($F_{(4, 28)} = 36.32$, $P < 0.001$). The freezing of naive and 15 kHz animals was similar ($P > 0.05$) and, in the meantime, it was lower than freezing of other groups ($P < 0.01$). Freezing of 7 kHz animals was different also from 3 kHz and 1 kHz (CS) group ($P < 0.001$). **c** Freezing in "discriminator" animals (D, $n = 5$) was lower than that observed in "generalizer" (G, $n = 4$) animals during the 7 kHz tone delivery ($P < 0.001$). **d** Time course of catFISH experiments. **e** Representative images showing neurons expressing single nuclear *H1a* (green arrows) and *Arc* (red) mRNA expression and double-labeled cells (yellow) in the naive, 15 kHz, 7 kHz (discriminators and generalizers), 3 kHz, and 1 kHz groups. Scale bar, 20 μm. **f** Dot plots showing the percentage of cells activated following new tone presentation (expressing only *H1a*), CS presentation (only *Arc*), and during both events (expressing both *Arc* and *H1a*). These results revealed an increase in *H1a* (new tone)- or *Arc* (CS)-expressing neurons and a decrease in double-labeled cells in the "discriminator" group ($F_{(5, 27)} = 13.68$ (left), $P < 0.001$; $F_{(5, 27)} = 12.68$, $P < 0.001$ (middle); $F_{(5, 27)} = 30.28$, $P < 0.001$ (right)). Raw data were expressed as a number of neurons labeled for *Arc*, *H1a* or both mRNA divided for the all counted neuronal nuclei analyzed. For each animal, we then calculated the mean of these raw data. **g** The total rate of *H1a* was lower in both the naive and the 15 kHz groups than in the other groups ($F_{(5, 27)} = 24.67$, $P < 0.001$). **h** The total rate of *Arc* was lower in the naive rats ($F_{(5, 27)} = 28.39$, $P < 0.001$). **i** Venn diagrams showing the percentage of *H1a*- (green), *Arc*- (red), and double- (yellow) labeled neurons in LA in the different experimental conditions. Diagrams' size was scaled on the basis of *H1a* or *Arc* total ratios, and percentages were calculated by dividing the number of *H1a*-, *Arc*-, and double-labeled neurons for the total number of cells activated in at least one of the two events. *$P < 0.05$, **$P < 0.01$, ***$P < 0.001$. All data are mean and SEM. One-way ANOVA with Newman–Keuls test (**b**, **f**, **i**, **j**)

To better investigate the differential activation of LA neurons, we calculated the reactivation ratio by dividing the amount of doubly labeled cells for the number of H1a-positive cells. Then, we compared this index with the reactivation ratio predicted by chance. The observed ratios were significantly higher with respect to predicted chance levels in each behavioral group (Supplementary Table 1)[24]. To depict the overall activity occurring in all groups, we calculated the percentage of neurons that were active

during the first (Fig. 1g) or the second (Fig. 1h) event. During the first event, Group 7 kHz G and 7 kHz D did not differ each other and neither from Group 1 and 2, whereas all of these groups differed from Group 4 and naive animals (Fig. 1g, i). During the second event, all groups differed from naive animals (Fig. 1h, i).

To address whether our results represent a general feature of LA participation to fear discrimination irrespective of the tone frequencies employed, we performed similar experiments but by

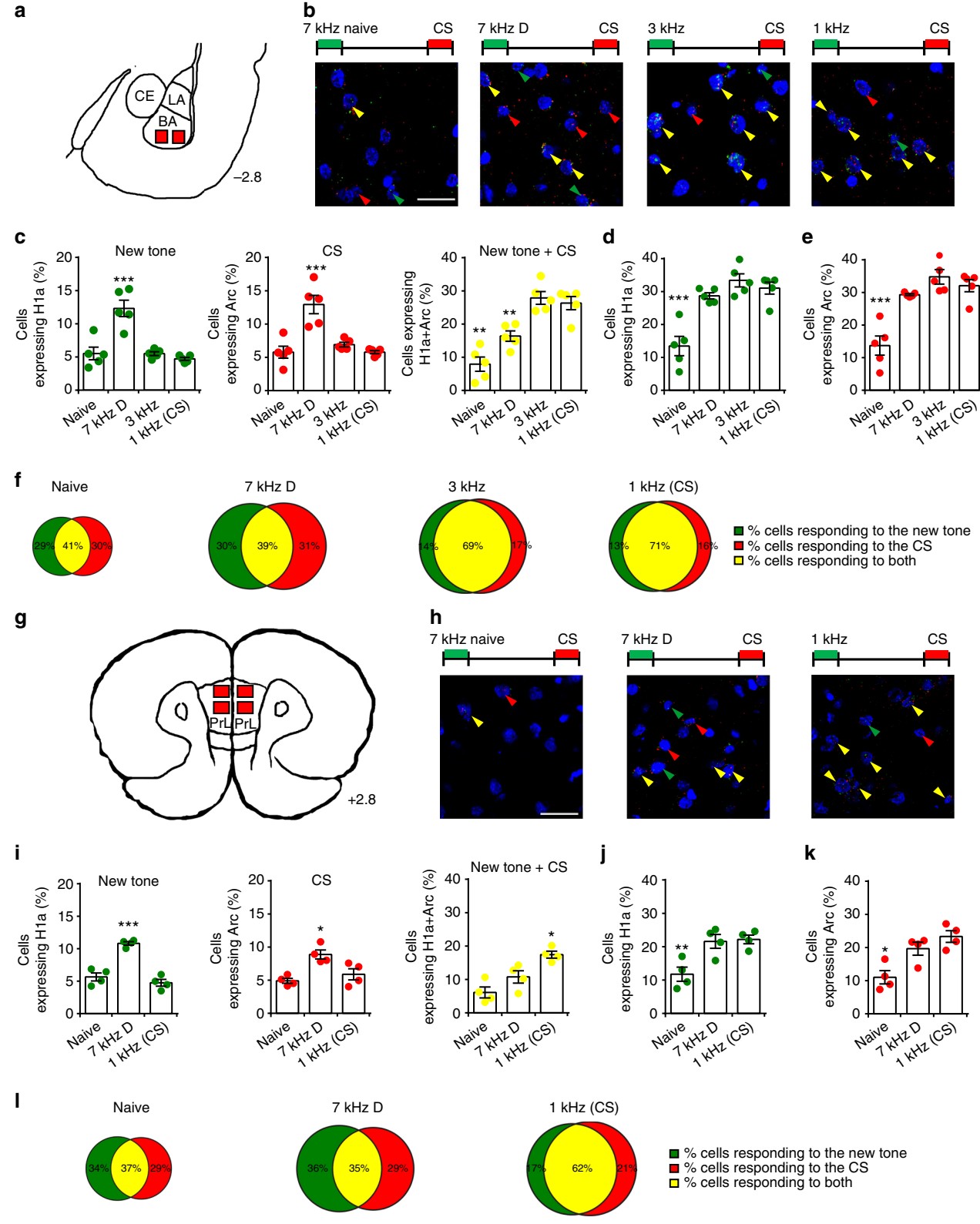

counterbalancing the tone paired to the US. Rats were conditioned to associate a 15 kHz tone to the US. One week after training, rats were presented with the CS (15 kHz) (Group 1) or with new tones of increasing different frequencies (7 kHz, Group 2; 1 kHz, Group 3). Twenty min later, all groups were presented with the CS to test their fear memory retention (Supplementary Fig. 5a). catFISH analysis revealed results similar to those detected in the above experiment (Supplementary Fig. 5).

Finally, we found similar neuronal processes in two other structures that are involved in fear discrimination, namely the basal amygdala (BA) (Fig. 2a–f) and the prelimbic area (PrL) (Fig. 2g–l) of the medial prefrontal cortex. These data suggest that the recruitment of different active neuronal subpopulations may be a common neuronal mechanism that occurs in several brain sites during discrimination processes.

**Characterization of the neurons activated by harmless cues.** We next addressed the question of which specific cell types are involved in discriminator animals using triple fluorescent in situ hybridization. We combined *catFISH* with the detection of a third riboprobe for a specific neuronal marker. By analyzing colocalization with CaMKIIa, a marker of excitatory neurons, we found that within the LA, the majority of the cells that responded to the CS were excitatory neurons (70.89 ± 2.45%). Strikingly, a large percentage of neurons (50.54 ± 5.64%) activated by the new tone also overlapped with the CaMKIIa-positive population (Fig. 3a, b and Supplementary Fig. 6). Thus, within LA, there are distinct subpopulations of pyramidal neurons that are engaged by threat CS or by new harmless stimuli. We then analyzed whether the excitatory neurons activated during discrimination processes belonged to a specific excitatory subpopulation. Previous studies showed that in the BA (but not in the LA), excitatory pyramidal neurons are engaged during the extinction processes to mediate fear inhibition (i.e., "fear off" neurons)[18,25]. Some of these neurons are identified by the expression of Thy1, which distinguishes a specific subpopulation of pyramidal neurons[25,26]. We therefore sought to determine whether the excitatory neurons activated during the discrimination process might express Thy1. Triple-FISH analysis revealed that during the presentation of the new tone, CS or both, only a minimal fraction of excitatory neurons expressed Thy-1 (Fig. 3c, d and Supplementary Fig. 6). These results are consistent with those of previous studies[25] demonstrating that Thy-1 is expressed within the BA but not in the LA.

A previous study showed that a gene, gastrin-related peptide *(Grp)*, is expressed within the pyramidal neurons of the LA[27]. These excitatory neurons release GRP peptide, which activates inhibitory neurons to increase the GABAergic inhibition of principal neurons[27]. Remarkably, GRPR-deficient mice showed

enhanced fear memory[27]. We therefore sought to determine whether some of the pyramidal neurons activated during discriminative processes express *Grp*. Triple-FISH revealed that during the presentation of the new tone, CS or both, a large majority of excitatory neurons also express *Grp* (Fig. 3e, f and Supplementary Fig. 6).

We next investigated which inhibitory interneurons could be recruited by these processes by analyzing the colocalization of *Homer 1a* and *Arc* with either *parvalbumin* (PV), *somatostastin* (SOM), or *calretinin* (CR) mRNA (Fig. 3g–l and Supplementary Fig. 6). Some PV, SOM, and CR interneurons were activated only during CS presentation, whereas a different portion of each neuronal subtype was recruited during only discriminative processes. Moreover, a portion of *Arc + H1a* expressing neurons also belonged to these different interneurons (Fig. 3g–l). These data reveal that threatening and harmless stimuli activated different subtypes of PV, SOM, and CR interneurons in the LA. Because a large variety of interneurons subtypes have been described within the amygdala[28,29], the observed differences may reflect this heterogeneity, and future studies will identify the specific cellular markers that characterize distinct inhibitory subtypes. These findings also showed that *Arc* was expressed in inhibitory neurons of LA, as previously reported in hippocampal, somatosensory, and striatal inhibitory neurons[30]. To further address this issue, we analyzed the colocalization between *Arc* and *Gad1* (a neuronal marker for inhibitory neurons) mRNA in our samples. catFISH analysis revealed that the expression of *Gad1* occurred in 23.35 ± 2.28% of LA neurons expressing *Arc* (Supplementary Fig. 7).

Taken together, our data suggest that discrimination processes within the LA engaged an intricate network of both pyramidal cells and inhibitory interneurons that are partially different from those activated by the CS (Fig. 3m).

**Blockade of LA neurons induced fear generalization.** Do neurons activated during discrimination processes play a causal role in detecting harmless stimuli? To answer this question, selectively manipulated neurons activated during a specific experience without affecting neighboring cells. We accomplished this using a pharmacogenetic approach in which c-fos-lacZ transgenic rats carried a transgene where c-fos promoter drives lacZ gene transcription, leading to β-galactosidase (β-gal) expression[21,31,32]. Induction occurs only in strongly activated neurons in which β-gal and Fos are coexpressed and not in neighboring non-activated or weakly activated cells[21,31,32]. These neurons can be inactivated 90 min after rats have performed a behavioral task by administering the prodrug Daun02: β-gal converts Daun02 into

**Fig. 2** catFISH analysis of BA and PrL revealed that different neuronal populations are activated by a new tone or CS presentation. **a** catFISH analysis in the basal amygdala (BA). The section diagram was drawn on the basis of our DAPI-stained sections. CE central amygdala, LA lateral amygdala. **b** Representative images showing single-labeled H1a (green arrows)- and Arc (red arrows)-expressing cells, and double-labeled cells (yellow arrows) in the naive (n = 5), 7 kHz D (discriminators, n = 5), 3 kHz (n = 5) and 1 kHz (n = 5) groups. Scale bar, 20 μm. **c** In BA, following new tone and CS presentation, the percentage of H1a- or Arc-expressing cells was higher in the 7 kHz D group than in other groups ($F_{(3, 16)} = 19.57$, $P < 0.001$ (left); $F_{(3, 16)} = 16.41$, $P < 0.001$ (middle); $F_{(3, 16)} = 23.67$, $P < 0.001$ (right)). **d** Total rates of H1a ($F_{(3, 16)} = 19.44$, $P < 0.001$) or **e** Arc ($F_{(3, 16)} = 20.78$, $P < 0.001$) were lower in the naive than in the other groups. **f** Scaled Venn diagrams showing the percentage of H1a (green), Arc (red), and H1a + Arc (yellow) expressing neurons in BA in the different behavioral groups. The neuronal populations activated during both new tone or CS presentation in 7 kHz D group were less overlapping with respect to other groups. **g** catFISH was performed in the layers II-III of the prelimbic cortex (PrL). The section diagram was drawn on the basis of our DAPI-stained sections. **h** Images showing H1a and Arc nuclear expression in the naive, 7 kHz D and 1 kHz groups. **i** In PrL, the percentages of cells single-labeled for H1a or Arc were significantly higher in the 7 kHz D group than in the naive and 1 kHz groups ($F_{(2, 9)} = 44.82$ (left), $P < 0.001$; $F_{(2, 9)} = 9.91$, $P < 0.01$ (middle)). Conversely, the percentage of double-labeled cells was lower in both the 7 kHz and the naive group than in the 1 kHz group ($F_{(2, 9)} = 13.55$, $P < 0.01$ (right)). **j** The total rates of both H1a ($F_{(2, 9)} = 9.76$, $P < 0.01$) and **k** Arc ($F_{(2, 9)} = 10.93$, $P < 0.01$) were lower in the naive group than in the 7 kHz D and 1 kHz groups. **l** In PrL, scaled Venn diagrams indicated that in 7 kHz D group the neuronal populations activated during new tone or CS presentation were less overlapped with respect to other groups. *$P < 0.05$, **$P < 0.01$, ***$P < 0.001$. All data are mean and SEM. One-way ANOVA with Newman–Keuls test (**c**, **d**, **e**, **h**, **i**, **j**)

daunorubicin, which induces apoptotic cell death in 3 days after injection.

First, we verified whether *Arc* mRNA activation colocalized with cFos expressing cells in our experimental protocol. catFISH analysis showed that 87.23% ± 1.25 of *Arc*-labeled neurons expressed also *cFos* mRNA (Supplementary Fig. 8), in line with previous studies showing that *c-fos*-expressing cells exhibit also

the activation of other activity-dependent genes such as *Arc*, *H1a*, and *zif268*[33,34].

Transgenic rats were exposed to the association between a tone (1 kHz) and US, as described in the above experiments. One week later, they were presented with a new tone of 7 kHz, and 90 min later, the discriminator animals were bilaterally infused with Vehicle or Daun02 in the LA (Fig. 4a). As a further control group,

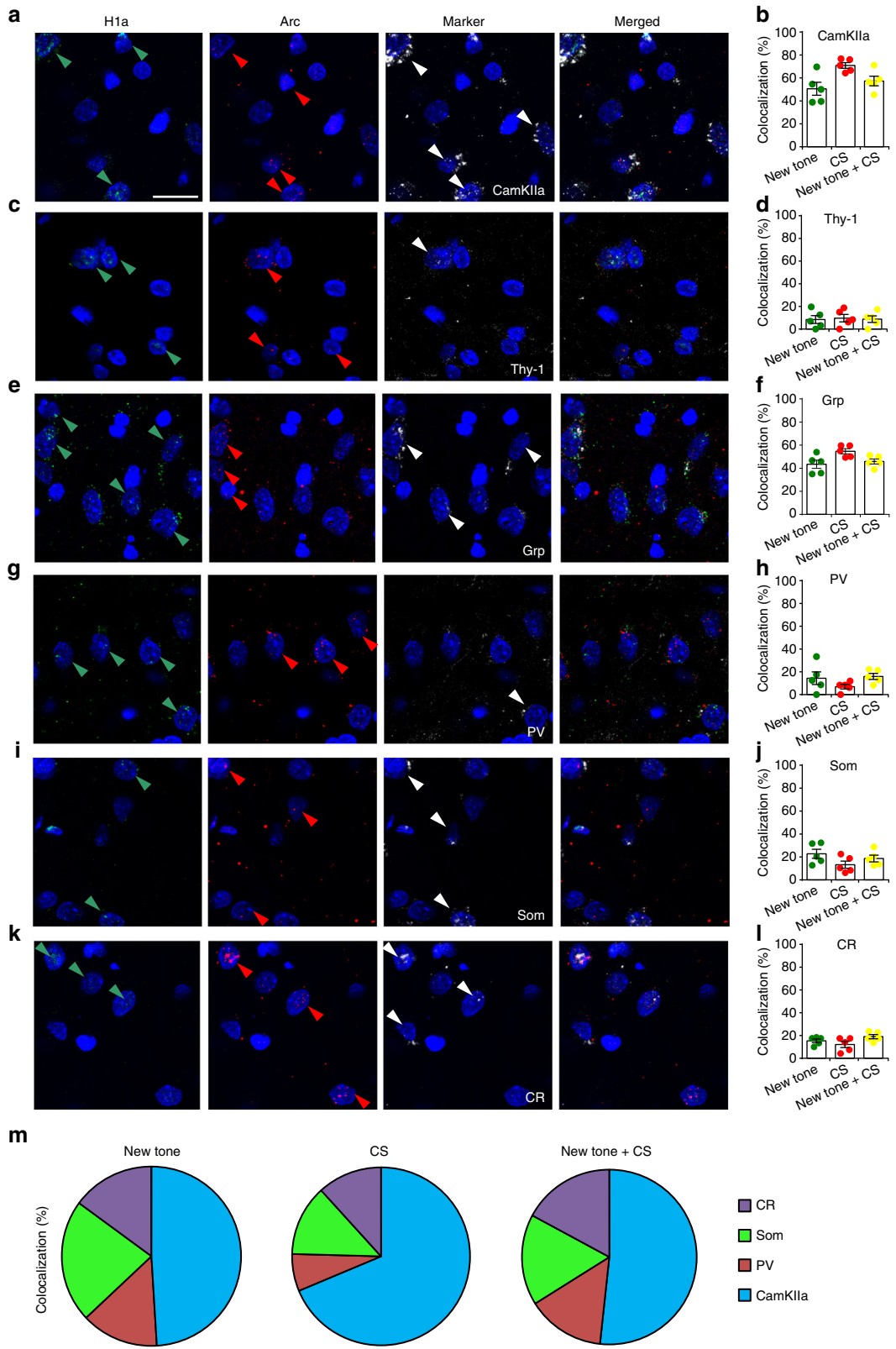

we added a "Daun02-delayed" group, in which Daun02 was injected one week after the new tone presentation, i.e., when the level of β-galactosidase expression induced by new tone was returned to baseline (Fig. 4a). On test day, the Daun02-injected animals displayed a significant increase in freezing in response to the 7 kHz tone with respect to controls ("Vehicle" and "Daun02-delayed" groups) (Fig. 4e and Supplementary Fig. 9), suggesting that the pharmacogenetic deletion of LA neurons activated during discrimination processes caused a shift from discrimination to fear generalization. A post-mortem cytochemical analysis revealed that the presentation of the new stimulus induced β-gal in 83.04 ± 2.25% of LA neurons expressing Fos (Fig. 4b) and that the Daun02-injected rats had markedly fewer Fos-expressing neurons (Fig. 4c, d).

It is unlikely that these results were due to nonspecific neurotoxicity in the LA or to interference with the overall functions of the LA because both conditions should have decreased (and not enhanced) defensive responses[35]. To further address this point, we tested fear memory expression by presenting the CS. No differences were detected among groups (Fig. 4e). In addition to confirming that our pharmacogenetic procedure did not affect the overall functions of the LA, the latter finding also indicated that neurons that are necessary for fear discrimination are conversely not necessary for CS memory expression. Intriguingly, catFISH analysis showed that some neurons were activated by both new stimuli and CS presentation. On the other hand, applying a pharmacogenetic approach after discrimination processes showed that these commonly activated cells are not necessary for fear memory expression. Thus, these neurons might be useful for comparing new stimuli with threatening ones, but they may be dispensable for fear memories. Future studies will help to define the precise role of these cells.

In the Vehicle- and Daun02-injected animals, we did not detect any difference in spontaneous behaviors displayed before CS presentation (Fig. 4e). To better define the effects of the pharmacogenetic deletion of specific LA neurons on spontaneous behavior, Vehicle- and Daun02-injected animals were submitted to the open field and elevated plus maze tests. In both paradigms, we did not detect any differences between groups (Fig. 4f, g). These data show that enhanced fear behavior was not the result of a general effect on anxiety-like or unconditioned fear responses.

We next investigated whether neurons activated during discrimination processes are specifically necessary for discriminative processes or whether they may serve to inhibit fear more generally and/or to switch from a high to a low fear state. Previous studies showed that neurons in the BA are necessary for this transition during fear extinction[18,25,26]. Therefore, in another group of animals, Daun02 or Vehicle was injected after the presentation of the new tone, and the animals subsequently underwent extinction training (Fig. 4h). No differences were found between groups during the extinction training. These data

reveal that neurons activated during discrimination between fear and harmless cues do not serve to inhibit fear nor to switch between exploratory and defensive behaviors. These findings are in line with the results of the catFISH analysis showing that neurons activated by discrimination belong to an excitatory subpopulation that is different from the "fear off" subpopulation[18,25,26]. Altogether, these data support the view that, in the LA, there are cells that are specifically involved in discriminating between harmless and threatening stimuli and that these cells are not required for extinguishing threatening stimuli.

To additionally test the specificity of our approach, in another group of transgenic rats, we administered Daun02 after CS presentation (Fig. 4i) and we found that it did not affect discrimination processes during the presentation of a new tone, whereas it impaired subsequent fear memory expression (Fig. 4i). These results confirm that different neuronal processes are engaged in the LA during discrimination process and fear memory expression and that the pharmacogenetic disruption of these different cell types produces opposite behavioral outcomes.

Our previous catFISH analysis also showed that neurons in the LA were specifically engaged in response to a 7 kHz tone but not to a 15 kHz tone. Therefore, another group of transgenic rats received Vehicle or Daun02 following the presentation of a new 15 kHz tone. On test day, the two groups showed similar responses to the 15 kHz tone and the CS (Fig. 4j). These data demonstrate that the LA is necessary for discrimination only in cases of uncertain stimuli to which an unambiguous interpretation is impossible, while it is dispensable if new stimuli are totally different from the CS.

## Discussion

Our findings shed new light on the neuronal processes that are engaged during the presentation of new stimuli. In keeping with previous studies[6,7], we found that LA activity is higher when new stimuli resemble threatening ones and lower when new stimuli largely differ from threats. On the other hand, we provide evidence showing that when the perceptual features of a new stimulus are neither close to nor very different from a threatening stimulus, this ambiguity may be resolved by the activation of two opposite neuronal subpopulations within the LA. In some animals, the same neurons that were activated by threats were also activated in the presence of the new stimulus, and this led to generalized fear. Conversely, in other rats, the same new stimulus led to the activation of neurons that are partially different from those activated during fear-related experiences. The recruitment of this neuronal ensemble allows discriminative processes to occur and produces an opposite behavioral outcome. Deleting the latter neuronal subpopulation caused fear generalization but did not affect innate fear, fear memory expression or extinction processes. These findings support the idea that these cells are

**Fig. 3** Neuronal characterization of different subpopulations activated in the discriminator rats following a new tone or CS presentation within the LA. **a** Representative photomicrograph of triple catFISH showing neurons expressing H1a (green arrows), Arc (red), and CamKIIa (white) mRNAs. The merged panel shows nuclei that were single-, double- or triple-labeled for H1a, Arc, and CamKIIa. Scale bar, 20 μm. **b** Dot-plots graphs showing the percentage of cells coexpressing either H1a (new tone) or Arc (CS) or H1a + Arc (new tone + CS) with CamKIIa. **c** Images of triple catFISH showing expression of H1a, Arc, and Thy-1 mRNAs. **d** The percentages of cells coexpressing Thy-1 and either H1a or Arc or those coexpressing H1a + Arc were low and similar among the three conditions. **e** Images showing the expression of Arc, H1a, and Grp mRNAs. **f** The percentages of cells coexpressing Grp and either H1a or Arc or both mRNAs were high and similar among the three conditions. **g** Photomicrographs showing neuronal expression of H1a, Arc, and Parvalbumin (PV) mRNAs. **h** The percentages of cells coexpressing PV were similar in neurons that were activated following the presentation of the new tone or the CS. **i** Images showing triple catFISH for Arc, H1a, and Somatostatin (SOM) mRNAs. **j** The percentages of cells coexpressing SOM with Arc, H1a, or H1a + Arc were similar following the presentation of the new tone or the CS. **k** Images of triple catFISH showing the mRNA expression of H1a, Arc, and Calretinin (CR). **l** There were no differences in the percentages of cells coexpressing CR with H1a, Arc, or H1a + Arc. **m** Pie charts summarizing the percentages of cells in different subpopulations of neurons that were activated by the new tone, CS or both, according to the expression of different neuronal markers. All data are mean and SEM

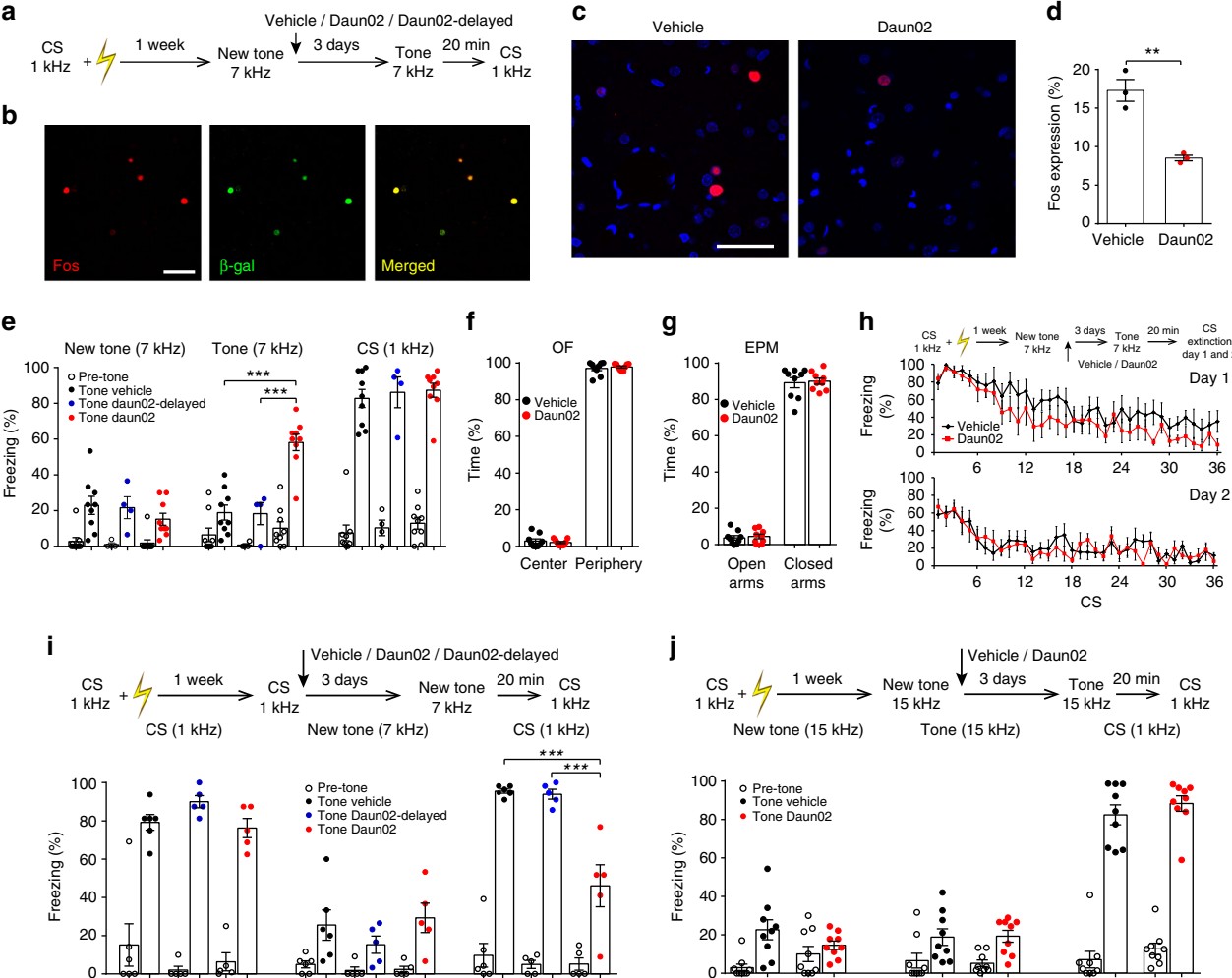

**Fig. 4** LA neurons activated by a new tone presentation are necessary for fear discrimination processes. **a** Experimental design for the Daun02 inactivation technique, see Methods for details. Daun02 ($n = 9$), Vehicle ($n = 9$), "Daun02-delayed" group ($n = 4$). **b** New tone presentation induced Fos (red nuclei) and β-galactosidase (green nuclei) expression in LA neurons. The merged panel shows nuclei that were double-labeled for β-galactosidase and Fos (yellow). Scale bar, 100 μm. **c** Photomicrographs of Fos-stained LA neurons in the Vehicle- (left) and Daun02-injected (right) rats at 90 min after 7 kHz sound presentation. Scale bar, 50 μm. **d** There were more Fos-positive cells in the Vehicle- than in the Daun02-injected animals. **e** Freezing response of the three groups ("Vehicle", "Daun02-delayed", and "Daun02") during new tone (7 kHz) presentation was similar among groups. After Daun02 injection, freezing to the 7 kHz tone was higher than in the two control groups (3 × 2 mixed ANOVA, main effect of group: $F_{(2, 19)} = 7.25$, $P < 0.01$, main effect of trial: $F_{(1, 19)} = 9.23$, $P < 0.01$, group × trial interaction: $F_{(2, 19)} = 20.07$, $P < 0.001$, simple main effect of group in post-injection trial: $F_{(2, 19)} = 24.13$, $P < 0.001$). Freezing to the CS was similar between groups (one-way ANOVA: $P > 0.05$). **f, g** In open field (OF) and elevated plus maze (EPM) tests, there were no differences between Vehicle and Daun02 groups. **h** No differences were detected during extinction sessions ($n = 6$ rats in each group). **i** In another experimental group, Daun02 ($n = 5$ rats), Daun02-delayed ($n = 5$), or Vehicle ($n = 6$ rats) was injected after CS presentation. Three days later, both a new tone and the CS were presented. Freezing responses to CS in the Daun02-injected animals were decreased (3 × 2 mixed ANOVA, main effect of group: $F_{(2, 13)} = 15.36$, $P < 0.001$, main effect of trial: $F_{(1, 13)} = 0.80$, $P > 0.05$, group × trial interaction: $F_{(2, 13)} = 14.11$, $P = 0.001$, simple main effect of group in post-injection trial: $F_{(2, 13)} = 20.57$, $P < 0.001$), but there were no differences between groups during new tone presentation (one-way ANOVA: $P > 0.05$). **j** Daun02 administration following the presentation of a new 15 kHz tone did not modify freezing to either this tone or to CS ($n = 9$ rats in each group). **\*\*$P <$ 0.01, \*\*\*$P <$ 0.001. All data are mean and SEM. Unpaired $t$ test (**d, j**); one-way ANOVA (**e, i**); 3 × 2 mixed ANOVA (**e, i**); 2 × 2 mixed ANOVA (**f, g, j**); 2 × 7 mixed ANOVA (**h**). See Supplementary Note 1 for a more detailed description of statistical results of this figure

specifically activated to recognize and react appropriately to new stimuli that are evaluated as harmless in uncertain situations rather than serving to switch fear off, as previously reported for some BA neurons[18,25,26]. They may therefore take part in resolving ambiguous situations.

Our data were obtained by testing rats one week after fear training. This allowed us to exclude that LA activity, when present, was due to any non-specific effects related to painful stimulation. Conversely, a previous study showed that *Arc* may be activated 15 min after painful stimulation irrespective of whether a fear memory was formed or not[36], thus suggesting that shortly

after training LA activity may be increased either by memory processes or by painful stimuli. A similar effect of painful stimulation on short-term neuronal activity was also reported in the cerebellar vermis[37].

Since Pavlov's studies[1], it has been known that behavioral responses towards stimuli that are increasingly dissimilar to one paired to an aversive event decay smoothly. The perceptual model of fear generalization proposes that neuronal activity arises in brain sites as a consequence of the perceptual similarity between a new stimuli and the CS[38,39]. Here, we show that the LA displays a fear-tuning profile when new stimuli either resemble the CS or

are totally different. Conversely, when a stimulus has intermediate features, there is no association between the number of recruited LA neurons and freezing levels. In this condition, the activity tuning within the amygdala may deviate from both perceptual features and the overt behavioral fear tuning. These data also suggest that imaging of the amygdala as a whole may not provide an accurate index of discriminative processes, at least in the LA.

It is likely that neurons activated within the LA may be part of a more complex network that encompasses several other brain areas, such as the medial prefrontal cortex[2,40,41] and the auditory cortex[20,42–45], both of which are involved in fear memory and discrimination processes.

Our pharmacogenetic experiments also demonstrate that inappropriate fear can be caused by disrupting the neurons that are engaged when animals encounter new harmless stimuli. This finding suggests a potential pathophysiological mechanism for the impaired discrimination that characterizes fear-related disorders, such as phobias and post-traumatic stress disorders. Stress and traumatic events might affect the correct functionality of these neurons, leading to fear overgeneralization. If confirmed, this finding suggests that appropriate treatments for fear-related disorders should not be aimed at decreasing amygdala global activity but should instead be directed toward strengthening the specific activation of these neurons.

## Methods

**Animals**. Sprague-Dawley male rats (age, 65–70 days; weight, 250–350 g) were employed all experiments except in those involving the blockade of Fos-activated neurons, where we employed *c-fos-lacZ* transgenic rats (age, 65–70 days; 250–350 g) that had been bred for 35–40 generations on a Sprague-Dawley background. Animals were housed in plastic cages with food and water available ad libitum, under a 12 h light/dark cycle (lights on at 7:00 a.m.) at a constant temperature of $22 \pm 1$ °C. All the experiments were conducted in accordance with the European Communities Council Directive 2010/63/EU and approved by the Italian Ministry of Health (authorization no. 322/2015).

**Fear conditioning**. Rats were trained to associate a conditioned auditory stimulus (CS) with a painful unconditioned stimulus (US) as in our previous studies[21,43,44]. The floor of the conditioning cage was made of stainless steel rods connected to a shock generator set to deliver 1 mA current. The chamber was fitted with a loudspeaker connected to a tone generator set to deliver an 80 dB, 1000 Hz pure tone (CS); the loudspeaker was located 20 cm above the floor. One animal at a time was placed inside the chamber and left undisturbed for 2 min. Then, it was exposed to a series of seven consecutive auditory CSs, each lasting 8 s and paired, during the last 1 s, with an electric foot shock (1 mA; 1 s); the seven sensory stimuli were separated by intervals of 22 s. "Familiar tone" group underwent the same experimental procedure but without any painful stimulation. Naïve animals were presented with the different tone only during the test trial.

**New auditory stimuli and fear memory retention test**. The presentation of novel auditory stimuli and the retention of fear memory was tested 1 week after the conditioning. Rats were handled for two consecutive days (5 min per day), habituated to an apparatus different from those used for conditioning and in a different room, in order to avoid conditioned fear behavior to contextual cues[43]. The cage consists in a transparent plastic cage enclosed within a sound-attenuating box equipped with an exhaust fan, which eliminated odorized air from the enclosure and provided background noise of 60 dB.

On the third day, we performed the behavioral test, divided in two different events separated each other by a time interval of 20 min in order to allow catFISH analysis. During the first event, after 1 min of free exploration, we presented a new auditory stimulus never presented before, repeated for four times (8 s, with a 22 s interval). In the separate behavioral groups, we delivered a tone of a different frequency (15 kHz, 70 dB; 7 kHz, 71 dB; 3 kHz, 73 dB). The session lasted 3 min after which the rat was returned to its home cage for 19 min. Then, animals were placed back in the same environment and exposed to CS (1 kHz, 72 dB) four times (8 s, with a 22 s interval). This sound was administered in a similar manner to that used during conditioning, but without the foot shock. A further behavioral group, was presented with 1 kHz sound during both events.

For the experiments in which we counterbalanced the frequencies of tones we repeated the same procedures except for the frequency of the CS (15 kHz) and of the new tones (7 kHz and 1 kHz).

The rats' behavior was recorded by a digital videocamera and the videos were reviewed to determine the duration of defensive responses. Freezing response was employed as an index of defensive behavior. Freezing was expressed as the percentage of time during which there was complete absence of somatic mobility, except for respiratory movements. The assessment of freezing was done by one person blinded to the animal's assignment to an experimental group.

In order to evaluate the rats belonging to the 7 kHz Group as discriminators "D" or generalizers "G", we employed an EM algorithm which yielded the maximum likelihood estimates for fitting a Gaussian mixture model (GMM)[46].

By applying this iterative method to the vector of freezing responses to the New 7 kHz Tone, under the assumption that data points were generated from a two-component mixture of Gaussian distributions, we obtained an approximation of the probability density that most likely generated the data. The EM-GMM estimated a threshold of ~43%. Therefore, animals with a freezing response lower than 43% to the New Tone were classified as discriminators, whereas animals showing a freezing higher than 43% to the New Tone were classified as generalizers (see Supplementary Fig. 2).

**Behavioral paradigms for Fos-LacZ experiments**. *Fos-LacZ* rats underwent fear conditioning as previously described (see Fear conditioning paragraph). One week after training, animals were presented with a new sound (7 kHz pure tone, 10 s). Each animal was placed inside the chamber and left undisturbed for 1 min. Then, it was exposed to the new tone, repeated for three times (10 s, with a 40 s of interval). Ninety minutes later, when β-galactosidase was near maximal levels[31,47] Daun02 or vehicle was bilaterally infused into the LA. Rats were returned to their home cages for 3 days in order to produce cell-specific inactivation[47]. On test day, both vehicle and Daun02-injected rats were returned in the cage and, after 1 min of exploration, 7 kHz tones were delivered, as previously described. Fear memory retention was tested 3 h later by delivering the CS (1 kHz pure tone) previously paired with the foot shock. "Daun02-delayed" rats underwent to the same experimental procedure but Daun02 injection was performed 1 week after new tone presentation.

In the second experiment, one week after fear conditioning, Daun02 was injected following the presentation of the CS (1 kHz pure tone, 8 s, interval of 22 s) previously paired with foot shock. The test was performed as in the first experiment.

The third experiment was performed as the first one (7 kHz experiment), but 15 kHz pips, lasting 1 s and delivered at 1 Hz for 15 s (inter-trial interval, 45 s), were used as new tone.

**Fear extinction protocol**. During the fear extinction procedure, *Fos-LacZ* rats were placed in the same environment as that in which they were presented with 7 kHz tone and were exposed to CS (1 kHz) 36 times (8 s, with a 32 s interval). This sound was administered in a manner identical to that used during conditioning, but without the foot shock. This paradigm was administered for two consecutive days, one session per day.

**Open field paradigm**. The open field apparatus consisted of a plastic opaque box (50 × 80 × 40 cm). Rats were placed in the center of the apparatus and their behavior was recorded for 10 min. The analyses were conducted using the Smart 3.0 software (Panlab, Cornella, Spain).

**Elevated plus maze paradigm**. The apparatus consisted of four arms (two open without walls and two enclosed by 30 cm high walls) 50 cm long and 10 cm wide. Each arm of the maze is attached to plastic legs, such that it is elevated 53 cm off a base that it is on. Rats were placed in the center of the apparatus and their behavior was recorded for 10 min. The analyses were conducted using the Smart 3.0 software (Panlab, Cornella, Spain).

**Stereotaxic surgery**. Stereotaxic coordinates for LA injections were taken from Paxinos and Watson[48] atlas. Two injections were performed bilaterally at the following coordinates: AP = 2.3, ML = ± 5.4, DV = 7.8 and AP = 3.3, ML = ± 5.4, DV = 8 to the Bregma. A burr hole, permitting the penetration of a 28 Gauge needle, was drilled over each injection site. The needle was connected to a 10 μl Hamilton syringe, connected to an infusion pump.

For the treatment of *c-fos-lacZ* transgenic rats, Daun02 (Hycultec, catalog no. HY-13061) was dissolved to a final concentration of 5 mg/ml in a solution of 10% DMSO, 6% Tween-80, and 84% phosphate-buffered saline. A volume of 1 μl of both Daun02 or vehicle was used per injection site (ν = 0.3 μl/min), and the needle was left in place for additional 3 min. The incision was then closed with stainless steel wound clips, and the animal was given a subcutaneous injection of the analgesic/anti-inflammatory ketoprofen (2 mg/kg body weight); it was kept warm and under observation until recovery from anesthesia. Needle track placement was verified in Nissl stained sections. The sections were histologically verified under a microscope magnified at ×2 and ×4.

**Tissue preparation and histological procedures**. Immediately after testing, rats were anesthetized and then rapidly decapitated with a guillotine. Brains were quickly removed and frozen in isopentane that had been supercooled on dry ice (approximately −80 °C). Frozen brains were stored at −80 °C. For sectioning, brains were warmed to −20 °C and fixed to the platform of a cryostat with Tissue-

TeK O.C.T. Compound (VWR). Sections (20 µm thick) were mounted on slides (Superfrost Plus, VWR), which were then sealed in boxes and stored at −80 °C until use.

**Fluorescent in situ hybridization (catFISH).** catFISH analysis was used to examine the expression of *Arc/Arg 3.1* (*Arc*) and *Homer 1a* (*H1a*) genes. Briefly, an *Arc* antisense riboprobe was directed to the region from exon I to III, while an *H1a* probe was directed to the 3′ UTR. The vectors were linearized with EcoR1, purified and used for in vitro transcription with the DIG RNA Labeling kit (SP6/T7) (Roche, 11175025910), in the presence of fluorescein-UTP (incorporated into the *H1a* probe) or digoxigenin-UTP (incorporated into the *Arc* probe). The yield and integrity of riboprobes was confirmed by gel electrophoresis. At the end of this process, probes were purified by spin chromatography.

To detect *Arc* expression, mounted sections were incubated with digoxigenin-labeled *Arc* riboprobe followed by anti-digoxigenin-POD (1:500, 11207733910, Roche) and a cyanine-3 substrate kit (1:50, NEL744001KT, PerkinElmer). After detection of the *Arc* riboprobe, the slideswere treated with 2% $H_2O_2$ to quench residual POD activity. Fluorescein-labeled *H1a* probe was detected with anti-fluorescein-POD (1:500, 11426346910, Roche) and a fluorescein substrate kit (1:50, NEL741001KT, PerkinElmer). Nuclei were counterstained with a mounting media containing DAPI (Vector, H1200). The specificity of the labeling was confirmed by omitting the riboprobes.

For characterization experiment, we add a third probe in order to detect the neuronal identity of *Arc* and *H1a* expressing neurons. To this aim, we analyzed the *CamKIIa*, *Thy-1*, *PV*, *SOM*, *CR*, and *GRP* mRNA expression. Primers used to make riboprobes were: for *CamKIIA*: (FW) ACCAACACCACCATCGAGGA, (RV) GGACGATCTGCCATTTTCCA; for *Thy-1*: (FW) GACCCAGGACGGAGCTATTG, (RV) TTTCTCCCCGCGTTTTGAGA; for *PV*: (FW) GCAGACTCCTTCGACCACAA, (RV) AGTCAGCGCCACTTAGCTTT; for *SOM*: (FW) CCCCAGACTCCGTCAGTTTC, (RV) AACGCAGGGTCTAGTTGAGC; for *CR*: (FW) GCACTTTGATGCTGACGGAA, (RV) GCCAAGGACATGACGCTCTT; for *Grp*: (FW) CAACGCACTCTCAGCCTAGT, (RV) GCTTCTTCCCAGCGGATGTA. These riboprobes were incorporated with biotin-UTP labeling mix (Roche, 11685597910). To detect them, following the detection of *H1a*, sections were incubated with anti-Biotin-POD (1:100, Vector, SP3010) and a Cy5 substrate kit (1:50, NEL745001KT, PerkinElmer).

In order to analyze a possible colocalization between *Arc* and *Gad1* or between *Arc* and *cFos*, the primers used to make riboprobes were: for *Gad1*: (FW) ACCAGATGTGTGCAGGCTAC, (RV) ACAGATCTTGACCCAACCTCTC; for *cFos*:(FW) TGTCAGGGGAAGAGTAGGGGTC, (RV) CCAGACACAGGTGGAGCAAG. These two riboprobes were incorporated with digoxigenin-UTP labeling mix (Roche), incubated with anti-digoxigenin-POD (Roche) and, then, with a Cy3 substrate kit (1:50, PerkinElmer). For this experiment*Arc* was incorporated in fluorescein-UTP labeling mix (Roche), incubated with anti-fluorescein-POD (Roche) and, then, with a Fluorescein substrate kit (1:50, PerkinElmer).

Slides were imaged using a Leica SP5 confocal microscope using four lasers (488, 520, 570, and 633 nm) corresponding to peaks in the emission spectra of DAPI (cell nuclei), fluorescein (*H1a* mRNA), Cy3 (*Arc* mRNA), and Cy5 (neuronal marker's mRNA), respectively. The objective lens was set at ×63 magnification. Data were acquired using a z-stack (1 µm thickness per section in a stack), the height of which was determined by the penetration of one detectable probe per sample (usually 8 µm thickness per stack). The pinhole, photomultiplier tube gain and contrast settings were constant for all image stacks acquired from a slide. Cells were considered for analysis if the nucleus was present in at least four sections of the z-stack. Only putative neurons were included in the analysis, and glial cells, identified from their small size (~5 µm diameter) and bright, uniform nuclear counterstaining, were excluded. Cells that were positive for both DAPI and Cy3 were considered *Arc*-positive, cells with both DAPI and fluorescein were considered *H1a*-positive, and cells with DAPI, Cy3, and fluorescein were positive for both mRNA. In triple catFISH analysis, we defined the number of cells expressing *H1a*, *Arc*, and *H1a* + *Arc* as described above. Then, we calculated the number of cells that were also Cy5-positive (neuronal marker's mRNA) in order to define the percentage of colocalization between *H1a*-, *Arc*- or double-labeled cells and each neuronal marker analyzed (*CamKIIa*, *Grp*, *Thy-1*, *PV*, *Som*, and *CR*). In double catFISH experiment performed BY using *Arc* and *Gad1* or *Arc* and *cFos* riboprobes, *Arc* was visualized in the emission spectra of fluorescein, and *Gad1* and *cFos* in the Cy3 emission spectra.

Cells counts were performed manually; to prevent bias, the experimenter was blinded to the relationship between the images and the behavioral conditions they represented. Raw data were expressed as a percentage of the total neuronal nuclei analyzed per stack. Typically, for PrL, 8 confocal z-stacks (175 × 175 µm square; zoom fraction, 1.4) were taken from cortical layer II–III of each animal: images were collected from two bilateral slides at 2.8 anteroposterior coordinate[48]. Thus, in this region we collected an average of 521 ± 30 DAPI-labeled cells per animal.

For LA and BA, four confocal z-stacks (189 × 189 µm square; zoom fraction, 1.3) in LA and eight confocal z-stacks in BA were taken from each animal: images were collected from two bilateral slides at anteroposterior coordinates ranging from

−2.6 to −3.0[48]. In LA, we collected an average of 174 ± 7 DAPI-labeled cells per animal, and in BA an average of 312 ± 15 DAPI-labeled cells per animal.

In order to define the percentage of colocalization between *Arc* and *Gad1* and *Arc* and *cFos*, the percentage of colocalization was obtained by counting the number of cells expressing *Arc* and *Gad1* or *Arc* and *cFos*.

In catFISH analysis, the percentages of stained cells for different groups were presented as mean and SEM.

**Beta-galactosidase and Fos immunohistochemistry.** On the injection day (Vehicle/Daun02), 90 min after testing, a group of *cfos-lacZ* rats was deeply anaesthetized and perfused intracardially with 4% paraformaldehyde in order to examine the colocalization of Fos and β-galactosidase protein expression. The brains were dissected, stored overnight at 4 ℃, and transferred to 30% sucrose. Coronal sections (50 µm) were cut on a vibratome and collected in phosphate-buffered saline (PBS). Free-floating sections were incubated in a blocking solution (4 % bovine serum albumin (BSA), 10% normal goat serum, and 0.5% Triton X-100) for 1 h at room temperature. Then, they were incubated in rabbit antibody to c-Fos (1:500 dilution, Santa Cruz Biotechnology, sc-52) and sheep antibody to β-gal (1:1000, Aves Labs, BGL-1010) in the blocking solution overnight at 4 ℃. Subsequently, sections were washed with PBS and incubated for 1 h at room temperature with AlexaFluor 488-labeled goat anti-sheep IgG (1:400 dilution, Life Technologies, A11039) and AlexaFluor-568-labeled goat anti-rabbit IgG (1:400 dilution, Life Rechnologies, A11036) diluted in PBS, for 1 h on a shaker at room temperature. Sections were washed in PBS, mounted with mounting media containing DAPI (Vector) and cover-slipped.

**Fos immunohistochemistry.** Three days after injection and 90 min after new tone (7 kHz) presentation, both vehicle and Daun02-injected rats were deeply anaesthetized and perfused intracardially with 4% PAF. Brain sections were processed for Fos immunohistochemistry. We used the primary antibody to c-Fos (1:500, Santa Cruz Biotechnology, sc-52) and the sections were developed with AlexaFluor-568-labeled goat anti-rabbit antibody (1:400 dilution, Lifetechnologies, A11036) as described in the previous section

**Immunohistochemistry analysis.** Tissues were imaged using three lasers (488, 520, and 570 nm), each corresponding this time to the peak emission spectrum for DAPI (Nissl stain for cell nuclei), Fluorescein *(β-gal)* and CY3 (*Fos*), respectively. Images were acquired using a z-stack (1 µm thickness per section in stack), the height of which was 8 µm. Cells were counted for analysis if the nucleus was present on at least 4 sections of the z-stack. The objective lens was set at ×63 magnification. Cells which were positive for both DAPI and CY3 were considered *Fos-positive*, cells with both DAPI and fluorescein were considered*β-gal*-positive, and cells with DAPI, CY3, and Fluorescein were considered double-labeled for both proteins. The results were expressed as a percentage of the total neuronal nuclei analyzed per stack.

Typically, four confocal z-stacks (189 × 189 µm square area; zoom fraction = 1.3) were taken in LA from each animal: images were collected from four slides at antero-posterior distance of −2.8 mm from the Bregma[48].

**Statistical analyses.** Since all data passed Kolmogorov–Smirnov's test and Brown–Forsythe test, parametric statistics were employed through all the experiments. F test was employed to test equality of variances where unpaired t test were used.

Concerning catFISH analysis, raw data were expressed as a number of neurons expressing *Arc*, *H1a* or both RNA divided for the total neuronal nuclei analyzed per stack. In order to determine the overall activity within LA during new tone or CS presentation we calculated the total rate of *H1a*, defined as the percentage of *H1a*-positive nuclei (expressing single *H1a* and both *H1a* + *Arc*) out of the total number of labeled nuclei. *Arc* total rate was calculated as percentage of *Arc*-positive nuclei (expressing single *Arc* and both *H1a* + *Arc*) out of labeled nuclei. The percentages of stained cells for different groups were presented as mean and SEM. In catFISH experiment, behavioral data and cell counts were analyzed by performing Student's two-tailed unpaired t test or one-way ANOVA followed by Newman–Keuls*post hoc* multiple comparison tests.

In order to test the main effect of group, the main effect of trial and the group × trial interaction effect in Fos-LacZ experiments, we employed a 3 × 2 and a 2 × 2 mixed-design ANOVA with group (Vehicle, Daun02-delayed, Daun02) as between-subjects variable and trial (pre-injection, post-injection) as within-subjects variable. Where the interaction was significant, we performed a simple main effects analysis (i.e., effect of group in pre- and post-injection trials separately).

In order to test the main effects of group and trial, and the interaction effect in the extinction paradigm, we employed three 2 × 7 mixed-design ANOVAs with group (Vehicle, Daun02) as between-subjects variable and trial (tones 1–7 in the Day-1 session, tones 30–36 in the Day-1 session, or tones 30–36 in the Day-2 session) as within-subjects variable. For each mixed ANOVA model we assessed the Sphericity assumption through Mauchly's Test of Sphericity. Where it was violated, we applied the Greenhouse-Geisser correction accordingly.

In order to compare the observed reactivation ratio and the reactivation ratio predicted by chance, we applied a calculation procedure[24]. We considered the

amount of green [G: $H1a + (H1a + Arc)$] and red [R: $Arc + (H1a + Arc)$] cells, and the total amount of neuronal nuclei in the same regions [D: DAPI]. We used Y/G to calculate the observed reactivation ratio, as in our previous work[21]. The chance for each neuron to be yellow [Y: $H1a + Arc$] is R/D*G/D, and the predicted Y = (R/D*G/D)*D = R/D*G. Thus, the predicted chance level of reactivation ratio is Y/G = (R/D*G)/G = R/D. The predicted ratio (R/D) and the observed ratio (Y/G) of each group were compared using a Student's two-tailed paired $t$ test (Supplementary Table 1). In order to test the difference between two groups, we used a Student's two-tailed unpaired $t$ test.

All statistical analyses were performed using Graphpad Prism 6 and SPSS Statistics 22 (IBM). The Gaussian mixture model was implemented with XLStat 19.4 (Addinsoft).

catFISH experiment was replicated two times while *Fos-LacZ* experiments were replicated three times.

**Data availability**. The data that support the findings of this study are available from the corresponding author upon reasonable request.

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

## Acknowledgements

We thank G. Merlo (Department of Molecular Biotechnologies and Health Sciences, University of Torino) for advices and the other members of our laboratory, M. Cambiaghi and G. Concina, for critical comments. We are grateful to Dr. P. Worley (Johns Hopkins University, Maryland, USA) for providing plasmids containing the *Arc* and *H1a* in situ probes and to Dr. T. Curran (Philadelphia Research Institute, Pennsylvania, USA) and Dr. B. Hope (Department of Health and Human Services, Maryland, USA) for providing cfos-lacZ transgenic rats. This work was supported by grants from the European Research Council (ERC) under the European Union's Seventh Framework Program (FP7/2007-20013)/ERC grant agreement no. 281072, the "Compagnia di San Paolo, Progetto d'Ateneo", University of Turin 2017 (CSTO167503), a Banca d'Italia

contribution and the Fondazione Giovanni Goria and Fondazione CRT (Talenti della Società Civile, ed. 2016).

## Author contributions

A.G., E.M., and A.R. devised, carried out, and analyzed behavioral experiments; A.G. and G.S. performed histological, catFISH, immunohistochemical, and confocal microscopy analyses; B.S. devises and analyzes the experiments and wrote the manuscript with input from A.G., G.S., and E.M. All authors discussed the results and commented on the manuscript.

## Additional information

**Competing interests:** The authors declare no competing interests.

