## [Peer Review File(PDF 285 kb) · Nature Communications]

Reviewers' comments:

Reviewer #1 (Remarks to the Author):

Authors examined that neural population involved in the auditory fear conditioning with specific frequency of tone (1kHz) is different from the population activated by the neutral tone with different frequency (7kHz, or 15kHz) in lateral amygdala. Authors tried to identify the cell-type of neural population activated by these stimuli. Finally, authors examined loss of function to understand the role of the population for fear memory expression and fear memory generalization. This is nice study to understand the functional significance of memory engrams in amygdala. While conceptually this is important study, there are a lot of missing data to proof their logic as a story.

Major:

- 1) As control experiments, in Figure 1, Authors should examine whether two types of tone (1kHz and 15 kHz (or 7kHz)) activate different population or not, without footshock conditioning.
- 2) Statistical analyses in their manuscript can be significantly improved. Authors need to calculate chance levels of double positive cells to make sure whether there is any significant difference between groups.
- 3) In Figure2 f-j, since PrL contains many different cell population which project differently (for example LII neurons preferentially project to amygdala), authors need to discriminate the each cells layers are activated by these stimuli.
- 4) As images shown in Figure 3 are very poor to convince readers to discriminate triple positive, double positive and single positive cells, authors need to improve the quality of pictures, show more examples in supplemental figs, and explain how they quantitatively define the positivity in methods.
- 5) In Figure 5a-e, authors should show proper control group which has Daun02 but without new tone (7kHz) exposure 1 week after conditioning, since c-Fos-LacZ mice cannot make any specific time window to label cells with LacZ. Similarly, In Figure 5i and j, Authors need proper control groups (e.g. Daun02 without CS 1 week after conditioning in Fig4i).

Minor:

It is better not to contain detail of material and methods in legends. The information has to move to methods section.

Reviewer #2 (Remarks to the Author):

The paper provides a very interesting study of how the amygdala processes generalized fear stimuli. It comes to the following key conclusions:

- 1) More BLA neurons are active during fear.
- 2) A unique population of neurons is activated when an intermediate test stimulus does not produce fear.
- 3) Neurons activated by a fear stimulus will reactivate when a similar stimulus also produces fear.

4) Lesioning the population of cells that activate when fear wouldn't generalize results in generalization—increased fear.

Only points 2 and 4 are novel. There may be an alternative interpretation of point 2 (see below). Point 4 is very novel and interesting and represents the strength of this paper. However, I do have a few methodological concerns that need to be addressed.

Point 2: A catFISH/fear study by Zelikowsky et al showed that there is a novel population of BLA cells activated on test regardless of whether or not conditioning has occurred. Those cells are not discriminating safety but it seems that the population reported here may be the same as those researchers found.

The design is somewhat problematic. Only one CS frequency was used (1kHz) and all generalization stimuli were higher in frequency. Some counterbalancing or at least a replication where the highest frequency was the CS would help be convincing that the experiments had generality. The low CS frequency is very near the bottom of the rat's hearing range and maybe a perceptually weird stimulus. So I am somewhat concerned that the results will not generalize to a different set of frequencies. Also, there is no information on whether or not presentation of the test frequencies was counterbalanced or not. This is especially important because the presentation of the test stimuli is quite massed and there may be pronounced carryover effects.

Vazdarjanova et al reported that in several brain regions Arc is not expressed in inhibitory neurons. I don't know that there is a similar analysis for amygdala, but expression in inhibitory neurons as suggested here is quite surprising and probably deserves additional corroboration.

Why did the B-gal studies use the fos rather than the Arc promoter? Expression of these two IEGs do not perfectly overlap. The analysis here assumes complete overlap.

Reviewer #3 (Remarks to the Author):

In this study Grosso et al. analyze the expression of immediate early genes (IEG) in response to a tone that was previously paired with a painful outcome, and in response to tones that were never presented before to the animal. The study uses the established techniques catFISH to quantify neural response in the lateral amygdala (LA) and pharmacogenetics inactivation to test the role of active neurons in this region. While the questions addressed are interesting, and the techniques are well chosen, the results are over interpreted and have major caveats. The authors make statements, already in the results section that are not supported by numbers. Moreover, data are poorly illustrated and many points remain unclear even after careful reading of the manuscript. This study needs major improvements to become suitable for publication in nature communication.

*** Major comments ***

First, in the Figure 1 it is unclear which data are represented for the cells counts (Figure1f-j).

1. from the methods or the legend it is not indicated whether the percentage of cells expressing H1a (left panel of Figure1f), the percentage of cells expressing Arc (second panel of Figure1f) are only expressing these markers or if they include the cells expressing both H1a and arc. From the numbers, it seems that they are representing the percentage of cells expressing only one of the markers, but since this is not clear the reliability of the data is questionable.

The author NEED to specify if the counts are H1a only and Arc only.

2. the authors should specify on the figure that the percentages are of all counted neuronal nuclei (large DAPI nuclei). This information is important and a sign of quality, but is buried in the methods.

The representation of the data is hard to decipher and the author did not use the appropriate way to represent the data. Pie charts or Venn diagrams would make the data much clearer and transparent.

p5: line 90: "As expected, in Group 1, the presentation of the two stimuli (CS-CS) engaged the same neurons (Fig. 1e, f)." This statement is wrong. Indeed, ~25 % of the cells expressing one or both IEG are expressing only one of them for the presentation of 1 kHz tone twice.

The authors do not counterbalance the tone that is predictive of the shock. Indeed it is known that rodents present preferential responsiveness to certain tone frequencies and this is a major confound for this study. The author should have used different tones frequencies as the shock predictor and compute the number of arc, H1a and arc+H1a cells depending on the frequency difference between the CS and the new tone. For example, in the Group 4 (15-1 kHz) the % of cells expressing Arc (in response to the CS) is much larger than in the other groups and it is unclear whether this increased response is due to the previous experience of another tone or to the tone frequency. Their experimental design does not allow to conclude on this essential point.

Moreover, across the manuscript the author use the 7 kHz-Discriminator subgroup as a reference. It is unclear why they did not chose, or include the 15 kHz group.

p6: the author state they "demonstrate that, when stimulus features are markedly different from the CS, LA activity is low, as previously proposed". Low is a very unspecific word and does not reflect that 20% of the neurons are expressing Arc after the 15 kHz stimulation (~4% Arc only + ~15% Arc+H1a).

The author do an important control with Naïve animals that did not undergo the fear conditioning. However, since the authors are interested in the association of the CS+US they should have done the fear conditioning session but unpair the CS-US, or at least, present only the CS.

Finally the quantification of neural activity and the inactivation of the LA active neurons uses different IEG which is a major confound and should be addressed.

*** Minor comments ***

Abstract: "dealing with" is an informal phrase. Please use more specific behavioral vocabulary.

Introduction

- line 39 - remove "reasons of"

Why only 4 z-stacks for the LA which is the major focus of the paper, and 8 z-stacks in BA and PFC ?

Title: "The neuronal basis of" suggests the circuit mechanism identified in this study is the only substrate of fear discrimination in the LA. Please temper the title by changing it to "A neuronal basis for".

Responses to reviewers

Responses to reviewer 1

Comment: As control experiments, in Figure 1, Authors should examine whether two types of tone (1kHz and 15 kHz (or 7kHz)) activate different population or not, without footshock conditioning.

Response: We thank the reviewer for this comment. We therefore modified Fig 1 by comparing Naïve animals and all other behavioral groups (see Fig. 1b, e and f). On the basis of Reviewer 3 suggestions, we also added a new experimental group (“Familiar Tone”), in which rats were exposed to 1 kHz tone unaccompanied by any US and, one week later, presented with both 1 and 7 kHz tones. As we did not detect any differences between this group and naïve animals (Supplementary Fig. 1), the two groups were collected altogether in Fig. 1.

Comment: Statistical analyses in their manuscript can be significantly improved. Authors need to calculate chance levels of double positive cells to make sure whether there is any significant difference between groups.

Response: We thank the reviewer for this suggestion. In order to investigate whether there is any significant difference between groups, we calculated the observed reactivation ratio and the reactivation ratio predicted by chance (see Methods). We found that the observed results were significantly higher with respect to predicted chance levels in all behavioral groups (Supplementary Table 1).

Comment: In Figure2 f-j, since PrL contains many different cell population which project differently (for example LII neurons preferentially project to amygdala), authors need to discriminate the each cells layers are activated by these stimuli.

Response: In the Method section we specified that catFISH analysis was performed in Layer II-III of the PrL.

Comment: As images shown in Figure 3 are very poor to convince readers to discriminate triple positive, double positive and single positive cells, authors need to improve the quality of pictures, show more examples in supplemental figs, and explain how they quantitatively define the positivity in methods.

Response: In order to better visualize *H1a*, *Arc* and neuronal marker' signals, for each triple catFISH, we included a picture for each channel analyzed (Fluorescein, Cy3 and Cy5) and the merged image (Fig. 3). Moreover, in order to show more examples of triple catFISH, a similar figure was inserted in which images were acquired from other fields of LA area (see Supplementary Fig.6). In each picture we used colored arrows to indicate the nuclei expressing *H1a*, *Arc* and each neuronal marker. In Material and Method section, we explained how the cell count was performed

(see p. 20).

Comment: In Figure 5a-e, authors should show proper control group which has Daun02 but without new tone (7kHz) exposure 1 week after conditioning, since c-Fos-LacZ mice cannot make any specific time window to label cells with LacZ. Similarly, In Figure 5i and j, Authors need proper control groups (e.g. Daun02 without CS 1 week after conditioning in Fig4i).

Response: We really thank the reviewer for this implementation of our study. Indeed, we performed a control group (“Daun02-delayed”), in which Daun02 was injected one week after the new tone presentation, i.e., when the level of β -galactosidase expression induced by new tone was returned to baseline (Fig. 4a-e). No differences were detected between this group and the “Vehicle” control group (Fig. 4a-e). We couldn’t inject Daun02 without new tone exposure because, before injection, we had to know whether the animal was “discriminator” or not. We also added the “Daun02-delayed” group in which Daun02 was injected one week after CS presentation. Neither discrimination processes nor fear memory expression were affected by this procedure (Fig. 4i).

Comment: It is better not to contain detail of material and methods in legends. The information has to move to methods section.

Response: We modified the Legends accordingly to the reviewer’s suggestion by removing some details of Materials and Methods from Legends.

Responses to reviewer 2

Comment: A catFISH/fear study by Zelikowsky et al showed that there is a novel population of BLA cells activated on test regardless of whether or not conditioning has occurred. Those cells are not discriminating safety but it seems that the population reported here may be the same as those researchers found.

Response: We thank the reviewer for raising this comment. We discussed this point in the Discussion section (p. 12).

Comment: The design is somewhat problematic. Only one CS frequency was used (1kHz) and all generalization stimuli were higher in frequency. Some counterbalancing or at least a replication where the highest frequency was the CS would help be convincing that the experiments had generality. The low CS frequency is very near the bottom of the rat’s hearing range and maybe a perceptually weird stimulus. So I am somewhat concerned that the results will not generalize to a different set of frequencies. Also, there is no information on whether or not presentation of the test frequencies was counterbalanced or not. This is especially important because the presentation of the test stimuli is quite massed and there may be pronounced carryover effects.

Response: We thank the reviewer for raising this critical point and for giving us useful suggestions to face it. We repeated our experiment but by counterbalancing the frequency of the tone paired to the US. In more details, rats were conditioned to associate a 15 kHz tone to the US. One week after training, they were presented with the CS (15 kHz) (Group 1) or with new tones of increasing different frequencies (7 kHz, Group 2; 1 kHz, Group 3). Then, all groups were presented with the CS (15 kHz). We obtained results similar to those presented in the early version of our study (see Supplementary Fig. 5). Combined, these data suggest that our findings represent a general feature of LA participation to fear discrimination irrespective of the tone frequencies employed.

Comment: Vazdarjanova et al reported that in several brain regions Arc is not expressed in inhibitory neurons. I don't know that there is a similar analysis for amygdala, but expression in inhibitory neurons as suggested here is quite surprising and probably deserves additional corroboration.

Response: In the literature, it has been observed that “some GAD65/67-positive neurons were observed to express Arc” in hippocampal, somatosensory and striatal regions (Vazdarjanova et al., 2006). However, since there is a lack of studies that had examined whether in LA Arc-positive neurons could display an inhibitory identity, we analyzed the colocalization between *Arc* and *Gad1* (a neuronal marker for inhibitory neurons) mRNA in our samples and we found the expression of *Gad1* in $23.35 \pm 2.28\%$ of LA neurons expressing *Arc* (see Supplementary Fig.7).

Comment: Why did the B-gal studies use the fos rather than the Arc promoter? Expression of these two IEGs do not perfectly overlap. The analysis here assumes complete overlap.

Response: We thank the reviewer for highlighting this point. Previous studies showed that in hippocampus and prefrontal cortex *c-fos*-expressing cells exhibit also the activation of other activity-dependent genes such as *Arc*, *H1a* and *zif268* (Fanous et al., 2013; Guzowski et al., 1999). To better address this issue, we verified whether in LA *Arc* mRNA activation colocalized with *cFos* expressing cells in our experimental condition. catFISH analysis showed that $87.23\% \pm 1.25$ of *Arc*-labeled neurons expressed also *cFos* mRNA (see Supplementary Fig. 8).

Responses to reviewer 3

Comment:

First, in the Figure 1 it is unclear which data are represented for the cells counts (Figure1f-j).

1. from the methods or the legend it is not indicated whether the percentage of cells expressing H1a (left panel of Figure1f), the percentage of cells expressing Arc (second panel of Figure1f) are only expressing these markers or if they include the cells expressing both H1a and arc.. From the numbers, it seems that they are representing the percentage of cells expressing only one of the

markers, but since this is not clear the reliability of the data is questionable.
The author NEED to specify if the counts are H1a only and Arc only.

Response: We thank the reviewer for this comment. To better clarify this point, we have changed the legend of Fig. 1 (panel f) accordingly.

2. the authors should specify on the figure that the percentages are of all counted neuronal nuclei (large DAPI nuclei). This information is important and a sign of quality, but is buried in the methods.

Response: We highlighted this point by adding this information in the legend of Fig. 1 (panel f)

Comment: The representation of the data is hard to decipher and the author did not use the appropriate way to represent the data. Pie charts or Venn diagrams would make the data much clearer and transparent.

Response: To better depict the neuronal activity in the different experimental conditions, we portrayed the percentage of *H1a*, *Arc* and *H1a+Arc* expressing neurons by using scaled Venn diagrams (see Fig. 1i; Fig. 2f, l; Supplementary Fig. 5). We thank the reviewer for this implementation of our data.

Comment: p5: line 90: “As expected, in Group 1, the presentation of the two stimuli (CS-CS) engaged the same neurons (Fig. 1e, f).” This statement is wrong. Indeed, ~25 % of the cells expressing one or both IEG are expressing only one of them for the presentation of 1 kHz tone twice.

Response: The reviewer is right and we changed the manuscript accordingly (p. 5).

Comment: The authors do not counterbalance the tone that is predictive of the shock. Indeed it is known that rodents present preferential responsiveness to certain tone frequencies and this is a major confound for this study. The author should have used different tones frequencies as the shock predictor and compute the number of arc, H1a and arc+H1a cells depending on the frequency difference between the CS and the new tone. For example, in the Group 4 (15-1 kHz) the % of cells expressing Arc (in response to the CS) is much larger than in the other groups and it is unclear whether this increased response is due to the previous experience of another tone or to the tone frequency. Their experimental design does not allow to conclude on this essential point.

Response: We thank the reviewer for raising this critical point. To address this important issue, we repeated the same experiment but by counterbalancing the tone that was associated to the footshock (US). Therefore, rats were conditioned to associate a 15 kHz tone to the US. One week after training, rats were presented with the CS (15 kHz) or with new tones of increasing different frequencies (7 kHz, 1 kHz) in the first event. 20 min later, all groups were presented with the CS (15 kHz). Through catFISH analysis we obtained results that were similar to the data already obtained and showed in Fig.1 (see Supplementary Fig. 5).

Comment: Moreover, across the manuscript the author use the 7 kHz-Discriminator subgroup as a reference. It is unclear why they did not chose, or include the 15 kHz group.

Response: We used the 7 kHz-Discriminator subgroup as a reference because only in this group we observed that discriminative processes engage a subset of neurons that are partially different from those engaged by fear processes. Instead, although 15 kHz group displayed similar behaviors during the presentation either CS or harmless stimuli as 7 kHz-discriminator animals, only the latter ones showed enhanced neuronal activity to either CS or harmless stimuli.

Comment: p6: the author state they “demonstrate that, when stimulus features are markedly different from the CS, LA activity is low, as previously proposed”. Low is a very unspecific word and does not reflect that 20% of the neurons are expressing Arc after the 15 kHz stimulation (~4% Arc only + ~15% Arc+H1a).

Response: We changed the manuscript accordingly (p. 6).

Comment: The author do an important control with Naïve animals that did not undergo the fear conditioning. However, since the authors are interested in the association of the CS+US they should have done the fear conditioning session but unpair the CS-US, or at least, present only the CS.

Response: We thank the reviewer for raising this important point. To address this issue, we introduced a new control group called “Familiar tone” in which rats were exposed to 1 kHz tone unaccompanied by any US and, one week later, they were exposed to either the 1 and the 7 kHz tones. catFISH analyses revealed that the LA activity was similar between “Familiar Tone” and “Naïve” groups (see Supplementary Fig. 1).

Comment: Finally the quantification of neural activity and the inactivation of the LA active neurons uses different IEG which is a major confound and should be addressed.

Response: In the literature it has been reported that in hippocampus and prefrontal cortex *c-fos*-expressing cells exhibit also the activation of other activity-dependent genes such as *Arc*, *H1a* and *zif268*, thus suggesting that these different immediate early-response genes are expressed in the same neuronal population (Fanous *et al.*, 2013; Guzowski *et al.*, 1999). Nevertheless, before adopting the “Daun02 inactivation method”, we verified whether in LA *Arc* mRNA activation colocalized with *cFos* expressing cells in our experimental condition. catFISH analysis showed that $87.23\% \pm 1.25$ of *Arc*-labeled neurons expressed also *cFos* mRNA (see Supplementary Fig. 8).

***** Minor comments *****

Comment: Abstract: “dealing with” is an informal phrase. Please use more specific behavioral vocabulary.

Response: We changed the Abstract as suggested.

Comment: Introduction - line 39 - remove “reasons of”

Response: We modified the manuscript accordingly.

Comment: Why only 4 z-stacks for the LA which is the major focus of the paper, and 8 z-stacks in BA and PFC ?

Response: In each of the 4 slices (for each animal) we decided to acquire 1 z-stack in the LA and 2 z-stacks in the BA and PrL regions because the whole extension of these regions is greater than that of LA (see Supplementary Fig. 4a and Fig. 2a, g). In this way we obtained a spatial reliable measure of the neuronal activation within these areas.

Comment: Title: “The neuronal basis of” suggests the circuit mechanism identified in this study is the only substrate of fear discrimination in the LA. Please temper the title by changing it to “A neuronal basis for”.

Response: We thank the reviewer for this suggestion and we changed the Title accordingly.

REVIEWERS' COMMENTS:

Reviewer #1 (Remarks to the Author):

Authors fixed all points which I mentioned. Now ready to move to next step.

Reviewer #2 (Remarks to the Author):

The authors added additional data to address my earlier concerns. I have only 2 minor points. I don't understand what is in the parentheses on page 7 lines 148-149. Maybe they meant decimal points not commas? That is the assumption I made when reviewing.

I don't see dB reported for each of the tones. This is especially important as many systems lose volume at higher frequencies.

The unique thing is the lesioning of the cells activated specifically when the animals discriminate and showing this promotes generalization. This is novel but not super surprising given the heterogeneous populations of cells already known in the BLA.

Reviewer #3 (Remarks to the Author):

The authors addressed the principal caveats of their study and included new analysis/representations as suggested. The manuscript has greatly improved from an ethical, and from a scientific communication standpoint. Their interesting findings are now suitable for publication.

Minor Comments

Line 116 : « in the latter case » = it is not clear what the authors are referring to.

Figure 1i, 2f, 2L : the legend is too small and barely readable. Also, a box surrounding the Ven diagram including the total number of counted neurons could facilitate understanding of the scaling.

Line 198 : 'Firstly' Because ordinal numbers (i.e., first, second, third, fourth, etc.) function as both adjectives and adverbs, the -ly adverbs firstly, secondly, thirdly, fourthly, and so on are superfluous. Also, this 'Fist' is not followed by a second or third.

Responses to reviewers

Responses to reviewer 1

Comment: Authors fixed all points which I mentioned. Now ready to move to next step.

Response: We thank the Reviewer for the useful comments. Through his suggestions we greatly improved the quality of the manuscript.

Responses to reviewer 2

Comment: The authors added additional data to address my earlier concerns. I have only 2 minor points. I don't understand what is in the parentheses on page 7 lines 148-149. Maybe they meant decimal points not commas? That is the assumption I made when reviewing.

Response: The reviewer is right. In order to correct this mistake we replaced commas with decimal points.

Comment: I don't see dB reported for each of the tones. This is especially important as many systems lose volume at higher frequencies.

Response: We thank the Reviewer for this comment. We reported dB for each of the tones in the Method section.

Responses to reviewer 3

Comment: Line 116 : « in the latter case » = it is not clear what the authors are referring to.

Response: We changed the manuscript in order to better specify what we are referring to.

Comment: Figure 1i, 2f, 2L : the legend is too small and barely readable. Also, a box surrounding the Ven diagram including the total number of counted neurons could facilitate understanding of the scaling.

Response: We thank the Reviewer for this comment. In order to facilitate understating of the scaling we modified Figure 1i, 2f and 2l by adding the percent sign in each Venn diagram. Moreover, in the legend of each figure, we better explained the data showed in Venn diagrams and how these diagrams were scaled.

Comment: Line 198 : ‘Firstly’ Because ordinal numbers (i.e., first, second, third, fourth, etc.) function as both adjectives and adverbs, the -ly adverbs firstly, secondly, thirdly, fourthly, and so on are superfluous. Also, this ‘Fist’ is not followed by a second or third.

Response: The reviewer is right and we changed the manuscript accordingly.